# Traits of Mortadella from Meat of Different Commercial Categories of Indigenous Dairy Cattle

**DOI:** 10.3390/ani14131980

**Published:** 2024-07-04

**Authors:** Cristina Giosuè, Giuseppe Maniaci, Riccardo Gannuscio, Marialetizia Ponte, Marianna Pipi, Antonino Di Grigoli, Adriana Bonanno, Marco Alabiso

**Affiliations:** 1Institute for Anthropic Impacts and Sustainability in the Marine Environment, National Council of Research (IAS-CNR), Lungomare Cristoforo Colombo 4521, 90149 Palermo, Italy; cristina.giosue@ias.cnr.it; 2Department of Agricultural, Food and Forest Sciences (SAAF), University of Palermo, Viale delle Scienze, 90128 Palermo, Italy; riccardo.gannuscio@unipa.it (R.G.); marialetizia.ponte@unipa.it (M.P.); marianna.pipi@unipa.it (M.P.); antonino.digrigoli@unipa.it (A.D.G.); adriana.bonanno@unipa.it (A.B.); marco.alabiso@unipa.it (M.A.)

**Keywords:** mortadella, cinisara cows, nebrodi black pigs, meat products, reformulation, fatty acid profile, sensory evaluation, autochthonous livestock breeds, market diversification

## Abstract

**Simple Summary:**

This study explores the production of Mortadella from Cinisara cow and young bull meats with pork lard, with animals primarily fed on natural resources. The investigation delves into the physicochemical, nutritional, and sensory aspects of Mortadella, aligning with the global interest in healthier meat products. Findings reveal a distinct fatty acid profile influenced by raw materials, with pistachios demonstrating a healthy fatty acid composition. Sensory evaluations underscore richer color, enhanced fat cube adhesion, and pronounced odors in cow-based formulations. While Mortadella formulations exhibit different nutritional profiles, the study emphasizes the impact of raw materials on these properties and highlights opportunities for interventions to improve fatty acid compositions in processed meat products. Overall, this research contributes valuable insights into producing healthier alternatives and supporting traditional farms, emphasizing regional diversity in the market.

**Abstract:**

The rising interest in healthier meat options prompted the exploration of alternatives to traditional pork-based products, incorporating meat from different livestock species, feeding regimens, and functional ingredients. This study investigates the production of healthier meat products by examining the physicochemical traits, fatty acid profile, and sensory properties of mortadella made with Cinisara meat of four young bulls and four adult cows, and four females of the Nebrodi Black Pig. All the animals were fed principally on natural resources. Nutritional analysis revealed different levels of moisture, protein, fat, and ash in raw materials, with pistachios contributing to a healthy fatty acid profile rich in monounsaturated and polyunsaturated fatty acids. Formulations using cow meat exhibited higher fat content and caloric value, resulting in sensory attributes such as more intense color, improved fat cube adhesion, and pronounced odors compared to young bull and control mortadella. Fatty acid analysis demonstrated distinctive profiles influenced by the meat type used and, as expected, bovine products showed higher contents of rumenic and other conjugated linoleic acids. Pork mortadella displayed greater ω6 and ω3 values, with a healthier ω6/ω3 ratio comparable to those found in cow products. Young bull mortadella showed the worse atherogenic and thrombogenic indices. The findings underscore the impact of raw materials on the nutritional and sensory attributes of mortadella, emphasizing the necessity for interventions to enhance fatty acid composition in processed meat products.

## 1. Introduction

The interest in producing innovative cured, fermented, and dried meat products, richer in protein and reduced in lipid content with a healthy fatty acid (FA) profile, is on the rise in accordance with WHO recommendations and evolving consumer preferences towards healthier options [1]. Traditional meat products such as salami and mortadella generally contain up to 30% fat, primarily sourced from pork, characterized by a prevalence of saturated and monounsaturated fatty acids [1,2,3]. The FA profile in fresh meat are influenced by factors like breed, age, gender, and feeding system [4,5,6]. Ruminants, through microbial activity in the rumen and diet-induced metabolic effects, transfer derivatives into meat and milk. Grazing, especially in ruminants, improves health and organoleptic properties, enhancing human-healthy FAs like ω3 and conjugated linoleic acid (CLA) [6,7].

Grazing influences also the meat flavor, producing a ‘grass fed’ taste in which other components of grass, such as indole compounds and terpenes, are also involved [8]. Otherwise, the products from animals that are concentrate-fed show generally high level of ω6, in particular linoleic acid, negatively affecting their healthy quality [9]. These compounds can therefore serve as markers of the animal’s feed background [10].

To obtain healthier animal products, recent studies explore alternative livestock (cattle, donkey, mutton, and poultry) with varied feeding systems and ingredients, yielding products with healthier nutritional profile. This presents economic opportunities, especially for farms in marginal areas with non-specialized breeds, diversifying market offerings [1,11,12,13,14,15,16,17,18,19,20,21].

In specific regions with autochthonous livestock breeds reared on natural resources, traditional farms face challenges in the fresh meat market. However, processed products, showcasing local specialties, can enhance economic profitability [22,23].

In this context, meat from Cinisara cow, a Sicilian autochthonous dairy breed [10,24], has shown promise in producing processed items like bresaola and salami [7,23], exhibiting physicochemical traits similar to other meat breeds. Furthermore, experimental tests conducted on calves raised on pasture and subsequently finished in the stable, according to the traditional breeding system, recorded valuable characteristics of the meat in terms of iron content, vitamin E, and the high concentration of conjugated linoleic acid (CLA). This is important in the prevention of cardiovascular diseases, while the lipid and cholesterol contents were significantly lower in muscle tissues compared to other breeds with a greater aptitude for meat production [22].

Together with salami, mortadella is another meat product traditionally obtained from an emulsion of the meat of one or more species of slaughter animals, including cows [25,26,27,28]. It may include the addition of pork fat and other ingredients, and it is typically encased in natural or artificial wrapping in various ways [29].

This product generally has high fat content, containing up to 30% fat, and presents unhealthy fatty acids characterized by prevalence of saturated and monounsaturated acids. To obtain Mortadella with a healthier nutritional profile, different studies have tested the use of alternative livestock such as cow, poultry, and donkey, and pork raised on natural resources, which, as is known, can improve the fatty acids profile in meat products by increasing polyunsaturated fatty acids content with important healthy effects for human consumers [11,12,13,14,15,16,17,18,19,20,21].

In this context, the present study provides information on the physicochemical traits, fatty acid profile, and sensory properties of lower fat content mortadella made with the meat of Cinisara young bulls and cows at the end of their productive life adding pork lard from the Nebrodi Black Pig, with the intent to also improve their fatty acid profiles, using animals raised on the prevalence use of natural resources.

## 2. Materials and Methods

### 2.1. Animals, Experimental Design and Mortadella Manufacturing

On a farm located in the typical breeding area of Cinisara cows, a total of 4 young bulls (18 months old) and 4 adult cows (10 years old) were selected based on age and sex. Pigs (4 females of the Nebrodi Black Pig—Sicilian autochthonous breed) from a farm situated in a representative area were chosen based on weight (approximately 110 kg). From 6 to 16 months of age, the young bulls were fed on pasture, while in the finishing phase (from 16 to 18 months of age), they were housed and fed with hay and concentrate. In contrast, the cows’ diet was primarily based on pasture and supplemented with hay and concentrate until slaughter. The pigs were bred outdoors and fed with concentrate until slaughter.

The animals were slaughtered at an EU-licensed abattoir, following standard handling procedures in accordance with EU regulations on the protection of animals at the time of slaughter [30].

The carcasses of all animals (4 cows, 4 young bulls, and 4 pigs) were stored in a cooling room at 4–8 °C for a 7-day aging period. Afterward, they were dissected, and the meat of each carcass was minced separately with a 6 mm plate. Subsequently, about 40 kg of mincemeat from each animal was minced using a 2 mm plate. Each batch of mincemeat was supplemented with pork lard from the Nebrodi Black Pig and whole pistachios (77.5% meat, 13% pork back-fat minced, 7.5% pork back-fat cubed, and 2% pistachio). Then, the following ingredients were added in the emulsion machine: 20% ice, 5% Protec MRT + Niko SA (nitrates and nitrites, ascorbic acid, milk powder and monosodium glutamate; Tec-AL s.r.l., Traversetolo, Italy), 0.1% Aromatec Mix Spez NRT 8672 NAT SA (mixture of spices and aromatic herbs; Tec-AL s.r.l., Traversetolo, Italy), and 0.04% Foxtec 201/1 (sodium and potassium diphosphates, sodium and potassium triphosphatates; Tec-AL s.r.l., Traversetolo, Italy). From each animal, 4 mortadella weighing about 12 kg were produced. The cooking cycle (time and temperature) was carried out according to an automatic cooking program: 65 °C (120 min) with a dry chamber, 75 °C (120 min) with a dry chamber, 80 °C (240 min) with a steam chamber, 75 °C with a dry chamber until a core temperature of 70 °C was reached. After cooking, the mortadella were showered with cold water and, after 60 min, placed in a blast chiller at −5 °C until the core temperature reached 0 °C; then, they were transferred to the refrigerator at 4 °C. Temperature control was monitored through interior sensors in chambers and on the mortadella.

All mortadella were produced at the “Lipari Salami Factory” in Alcamo (Sicily, Italy).

### 2.2. Sampling and Analysis

During the mortadella making process, minced meat (12 samples), complete mixture before cooking (12 samples), mortadella after cooking (48 samples), as well as pork lard and pistachio (in triplicate) were sampled, placed into sterile containers, and immediately refrigerated at −20 °C. They were then freeze-dried (SCANVAC Coolsafe 55-9, Labogene Aps, Lynge, Denmark) for successive analysis.

Each mortadella was weighed before and after cooking, expressing the weight loss as a percentage of the initial weight.

On fresh samples of raw materials, mixture pre-cooking, and mortadella, pH, water activity (a_w_), and colorimetric parameters were determined. The pH was measured with a digital pH meter (Thermo Orion 710 A+, Cambridgeshire, UK), equipped with a penetration probe.

The a_w_ was measured with a dew-point hygrometer HygroLab 3 (Rotronic, Huntington, New York, NY, USA) calibrated with five saturated solutions of known a_w_.

Colorimetric parameters were measured with a Chroma Meter (CR-300, Minolta, Osaka, Japan) using illuminant C, calibrated on a white standard such as L* = 100 (equivalent to BaSO_4_); results were expressed as lightness (L*, range from 0 (black) to 100 (white)), redness (a*, range from red (+a) to green (−a)), and yellowness (b*, range from yellow (+b) to blue (−b)), following the CIE L*a*b* system of International Commission on Illumination [31]. Chroma [C = (a*^2^ + b*^2^)^0.5^] and Hue [H = (arctg b*/a*)*57.296] were calculated with of a* and b* values.

The hardness of fresh mortadella was expressed as the maximum resistance to compression (compressive stress, N/mm^2^) determined using the Instron 5564 tester (Instron, Trezzano sul Naviglio, Milano, Italy) on samples (2 cm × 2 cm × 2 cm) kept at room temperature (22 °C). Samples were subjected to a compression with a load cell Lepetit [32] of 50 kg modified (4 cm^2^ square probe and cell with two lateral faces of 2 cm × 2 cm), run of 1.6 cm and speed of the crossbar 100 mm/min, recording maximum compression at the end of the run.

On freeze-dried samples of raw materials, mixture pre-cooking, and mortadella, moisture (method 967.03), crude protein (CP, N × 6.25) (method 988.05), ether extract (EE, method 920.29), and ash (method 942.05) contents were determined [33]. The total carbohydrate content was estimated by difference. Moreover, the caloric value (Kcal/100 g) was calculated considering the following conversion factors: 4.0 kcal/g for protein, 4.0 kcal/g for carbohydrates, and 9.0 kcal/g for lipids [34]. The oxidation status of fat was assessed by determining the peroxide value (POV, mEq O_2_/kg fat) as an index of primary lipid oxidation, as reported by Gaglio et al. [35]. On freeze-dried samples of raw materials and mortadella, the fatty acids (FA) were extracted according to the method developed by O’Fallon et al. [36], using C23:0 (Sigma-Aldrich, Darmstadt, Germany) as an internal standard (0.5 mg/g freeze-dried sample) for total FA quantification. Each sample (1 µL) was injected by autosampler into an HP 6890 gas chromatography system equipped with a flame ionization detector (Agilent Technologies Inc., Santa Clara, CA, USA), as described by Alabiso et al. [7].

The index of atherogenicity (IA) and the index of thrombogenicity (IT) were calculated according to Ulbritch and Southgate [37], as below reported:$$\mathrm{IA}=\frac{\left(C12:0+\left(4*C14:0\right)+C16:0\right)}{PUFA+MUFA}\;\mathrm{IT}=\frac{C14:0+C16:0+C18:0}{\left(0.5\cdot \Sigma MUFA)+(0.5\cdot \Sigma PUFA\omega 6)+(3\cdot \Sigma PUFA\omega 3)+(\Sigma PUFA\omega 3/\Sigma PUFA\omega 6\right)}$$

#### Sensory Evaluation

For the descriptive sensory analysis, 10 experienced panelists were recruited based on their prior participation in sensory panels for similar products. Additionally, a preparatory session was conducted before testing following the ISO 8589/2007 indications, allowing the panelists to thoroughly clarify and calibrate each attribute to be evaluated in mortadella.

One slice (1 mm thick with a 25 cm diameter) of each product was served one by one in a randomized order to each panelist. Samples were presented at a temperature of approximately 12 °C in hermetic transparent containers. Evaluations were conducted under regular white light. Water and unsalted crackers were provided to panelists to cleanse the palate between samples [27].

Panelists were instructed to assess odor descriptors (odor intensity, odor persistence, spiciness, unpleasant odors), visual characteristics (color intensity, color homogeneity, adhesion of fat cubes, presence of gelatin pockets), taste (salty, sweet, sour, bitter, spicy), tactile sensations in the mouth (tenderness, chewability, fatness), and overall acceptability of the product using a hedonic scale of 10 cm divided into 10 hedonic points (0 = absent; 10 = intense).

### 2.3. Statistical Analysis

The data for physicochemical parameters, fatty acid profiles, and sensory analysis were statistically analyzed using SAS 9.2 software [38]. A generalized linear model (GLM) was employed, which included the effects of the type of meat (3 levels, cow, young bull and pork). Results are reported as least squares means (LSM), and differences between means were assessed using Tukey’s *t*-test. Statistical significance was attributed to *p*-values ≤ 0.05.

## 3. Result and Discussions

### 3.1. Chemical and Physical Composition

During the production process, each type of meat (Table 1, Table 2 and Table 3) exhibited pH values ideal for the transformation into Mortadella, achieving pH values (5.74–6.16) slightly lower than those typically observed in this product (6.00–6.71) [14,15,19,21,28,29].

The concept of water activity (a_w_) has been widely used in food processing and preservation, correlating it with microbial growth, chemical reactions, and physical changes [38]. All mixtures (Table 2) showed aw values (0.84–0.87) that do not allow the growth of pathogenic bacteria but are favorable for yeast and molds (aw limit 0.80) [39]. All Mortadella samples (Table 3) presented aw values (0.94) in line with the product (<0.98), regardless of the meat type used in the mixture [21,29]. These results confirm that products like Mortadella require greater hygienic-sanitary control from the raw matrix to the final product to meet microbiological safety standards at the end of the manufacturing process [40].

Regarding nutritional aspects, Pistachio (Table 1) exhibited contents in moisture, protein, fat, and ash in line with those found by other authors in the same matrices [41]. On the other hand, cow and young bull meat, as well as pork, presented a chemical composition (particularly protein and fat) similar to those reported by USDA [42] for beef (90% meat + 10% fat) and raw pork boneless, respectively (Table 1).

The results of the chemical composition of the pork mortadella were in agreement with those found by other authors in similar product [29,43,44]. Considering the mixtures (Table 2) and the Mortadella products (Table 3), those containing pork meat presented a higher moisture (%) than those including young bull or cow, resulting in a lower fat and protein contents. All experimental Mortadella showed protein and ash content in line with those found by other authors in similar products, with lower moisture and greater fat contents [13].

The higher fat content in cow Mortadella resulted in a greater caloric value (Kcal/100 g) in this product compared to those made with young bull and pork (Table 3), and only pork Mortadella showed caloric content lower than the usual range for this product, which is 288 to 311 Kcal/100 g [42].

The peroxide values (POV) (Table 1, Table 2 and Table 3) were always <5, indicating that the fat did not turn rancid during the manufacturing process [45].

Considering the instrumental color, as expected, the lightness (L*) and redness (a*, related to the oxygenation of myoglobin) in mixtures and the final product based on pork were higher and lower than those found in young bull and cow (Table 2 and Table 3), respectively. In the present study, the yellowness (b*) was lower in the mixture and final product based on pork than on cow and young bull meat (Table 2 and Table 3). Since yellowness (b*) depends principally on the dietary transfer of carotenoids from pasture to meat fat, the amount of fat accumulated during finishing, and the rate of utilization of carotene from body fat [46], grazing feeding would be directly responsible of the higher values in cow and young bull products. The L* and the a* observed in cow and in young bull mortadella were comparable to those found by other authors in similar products, with only slightly lower b* [26,28]. The hue and chroma components of meat color are dominated by the pigment myoglobin, which changes color depending on its biochemical state, especially the degree of oxidation or reduction of myoglobin [47]. Our results indicated a more vivid color in the Mortadella formulation containing cow and young bull than in those including pork, probably due to the high content of heme pigments (Table 2). The final products maintained a certain difference in the vividness of the color, even if the hue was similar (Table 3). Considering the hardness, the cow mortadella showed higher resistance to compression than in young bull and control products.

### 3.2. Fatty Acids Composition

Fatty acids (FAs) play fundamental roles in the human body, serving as structural components of membrane lipids and precursors to eicosanoids and prostaglandins, and are essential components of metabolism [48]. The quantity and quality of FAs, including saturated fatty acids (SFAs), monounsaturated fatty acids (MUFAs), polyunsaturated fatty acids (PUFAs), ω-3 and ω-6 fatty acids, PUFA/SFA and ω-6/ω-3 ratios, and indices of atherogenicity and thrombogenicity, are important factors to consider in the nutritional evaluation of foods [48].

Examining the fatty acids in raw materials (% FA) (Table 4), Pistachios (*Pistacia vera* L.) exhibited a profile in line with observations from other studies [49], rich in MUFAs (68.8%) and PUFAs (17.5%), with a prevalence of oleic acid (18:1ω9; 68.6%), known for its hypoglycemic, hypocholesterolemic, cardioprotective, and anti-inflammatory properties. The high presence of unsaturated fatty acids along with low contents of SFAs (13.7%) renders pistachio a healthy food. Cow and pork fat showed contents in SFAs, MUFAs and PUFAs in line with those observed in other studies involving the Cinisara cow and the Sicilian Black Pig [50,51]. In contrast, young bull meat was higher in SFAs and MUFAs and lower in PUFA than meat of young bulls with a comparable feeding system [51], and showed a similar FA profile to those observed on beef and heifers crosses with continental beef breeds [52]. Unlike the preparation of Mortadella, the fat on the surface was removed for producing salami, as described by Alabiso et al. [51]. This fat is particularly rich in SFAs and MUFAs [53], and its removal could generate differences in the fatty acids profiles. The FA profiles in Mortadella (Table 5, Table 6, Table 7 and Table 8) reflected those observed in the raw ingredients (Table 4) and were similar to those observed in Mortadella-type products [1,54], showing consistent amounts of palmitic (C16:0; Table 5), stearic (C18:0; Table 5), oleic (C18:1 c9; Table 6), and linoleic (C18:2 ω6; Table 7) acids.

Considering SFAs (Table 5), cow and young bull Mortadella showed lower contents of short- and medium-chain fatty acids (from C8:0 to C12:0) than pork Mortadella, except for myristic acid (C14:0), which was higher in these products. An important amount of pentadecanoic acid (C15:0) was found in young bull Mortadella, which also exhibited the highest value of palmitic acid (C16:0). No differences were observed in the content of stearic acid (C18:0), while arachidic acid (C20:0) was detected only in pork Mortadella. Among SFAs, myristic acid, unlike palmitic acid, is considered more undesirable due to its significant hypercholesterolemic effect on human health [55]. Moreover, stearic acid can be transformed into oleic acid in body tissues, both beneficial for their hypocholesterolemic action [56].

Observing MUFAs (Table 6), these were higher in cow and young bull products from C12:1 to C17:1 than in control Mortadella, which instead showed the higher amount of oleic acid (C18:1 c9), generally the most representative FA in meat and meat products [7,51]. As expected, trans-vaccenic (C18:1 t11) acids were detected higher in beef products, being associated with the activity of microflora in the rumen.

Referring to PUFAs (Table 7), all products showed higher amount of linoleic acid (C18:2 ω6), while young bull and cow mortadella presented greater level of rumenic acid (CLA C18:2 c9t11 RA) and other CLA, in line with to those found in bovine meat and meat products [7,51].

Lower amounts of α-linolenic (C18:3 ω3), docosapentaenoic (C22:5 ω3), and docosahexaenoic (C22:6 ω3) acids were detected in cow and young bull products than in pork mortadella.

Young bull Mortadella showed intermediate values of γ-linolenic (C18:3 ω6) and arachidonic (C20:4 ω6) acids but a higher amount of eicosapentaenoic (C20:5 ω3) than those observed in cow and pork products.

As is known, the fatty acid profile in animal products is greatly influenced by the feeding system, and products obtained from animals grazing fresh forage on pasture generally exhibit profiles that are considered healthy for humans. This includes an increase in polyunsaturated fatty acid (PUFA) contents, ω3 fatty acids (which, along with ω6, cannot be synthesized by the human body and must be obtained through the diet), and conjugated linoleic acid (CLA), especially in ruminants [57]. SFAs tend to elevate low-density lipoproteins (LDL), leading to higher levels of cholesterol in the blood and an associated higher risk of cardiovascular disease [1]. In contrast, unsaturated fatty acids play a key role in reducing inflammation, increasing high-density lipoproteins (HDL), and reducing LDL [1]. In this context, various indices evaluate the quality of the lipid fraction and the health value of dietary fat, including the ω6/ω3 ratio and those based on PUFA amount in relation to SFA, as the index of atherogenicity (IA), and the index of thrombogenicity (IT).

Observing the results (Table 8), beef products showed lower amounts of ω6 and ω3 fatty acids than pork mortadella, but all products showed unfavorable PUFA/SFA and ω6/ω3 ratios for human health. According to FAO/WHO recommendations, a diet aiming to counteract various “lifestyle diseases” (such as coronary heart diseases and cancers) requires a PUFA/SFA ratio above approximately 0.45 and an ω6/ω3 ratio below 4–5 [58,59]. Considering the AI and TI indices, lower values are assumed to be more beneficial to human health, although no organization has yet provided recommended limits [60].

Yong bull mortadella showed the highest AI and TI values, that in all products were similar to those found by other authors in meats and meat products (AI between 0.165 to 1.320; TI between 0.288 to 1.694 [60].

The unbalanced fatty acid composition found is common to many delicatessen products. In fact, numerous studies have been conducted with the aim of improving the fatty acid composition of the lipid fraction of delicatessen products. This includes interventions at the zootechnical level with diets rich in sources of polyunsaturated fatty acids or their precursors, as well as the use of natural antioxidant compounds to reduce the oxidative degradation of the fat fraction rich in polyunsaturated fatty acids [61,62].

### 3.3. Sensorial Profile

Considering the sensorial profile (Table 9), Mortadella made with cow and young bull exhibited higher color intensity and homogeneity, as well as better adhesion of fat cubes. The odor was more persistent and spicier in cow and young bull products, respectively, while the control Mortadella had a less favorable smell. The cow and young bull Mortadella were perceived as sourer than the control one. Overall, all the products were well appreciated.

## 4. Conclusions

The findings of this study shed light on the intricate details of the productive process and the quality attributes of Mortadella formulations incorporating various meats. Notably, the cow Mortadella exhibited a higher fat content and caloric value compared to young bull and pork products. Instrumental color analysis revealed distinctive characteristics, with pork formulations displaying higher lightness and lower redness than those made with cow and young bull meats. Referring to the fatty acid profile of Mortadella products, among saturated fatty acids, cow and young bull Mortadella displayed lower contents of short- and medium-chain fatty acids compared to the control, with myristic acid being an exception.

Monounsaturated FAs exhibited variations, with cow and young bull products demonstrating higher levels of specific MUFAs (such as trans-vaccenic acid) than the control Mortadella, which instead showed the highest content of oleic acid.

Polyunsaturated FAs showed diverse patterns, with bovine Mortadella exhibiting amount of rumenic acid and other CLA in line with those found in similar products. Bovine products showed lower amounts of specific ω3 FAs, such as α-linolenic, docosapentaenoic, and docosahexaenoic acids.

Despite the potential health benefits associated with unsaturated fatty acids, all Mortadella formulations displayed unfavorable PUFA/SFA and ω6/ω3 ratios, highlighting the need to improve. All products showed AI and TI values in line with those generally found in meats and meat products.

Considering the sensorial profile, the Mortadella made with cow and young bull were sourer than the control one.

In conclusion, the unbalanced fatty acid composition found in Mortadella formulations underscores the challenges in achieving optimal nutritional profiles in delicatessen products. While these findings align with common trends in similar products, future studies could explore interventions at the zootechnical level and the use of other ingredients and antioxidants to enhance the nutritional quality of such food items. In this context, other studies could be directed towards improving technological processes to make Mortadella using only bovine meat and fat. This would also address the market demand from non-pork consumers, such as Muslims. Addressing these aspects could contribute to the development of healthier and more nutritionally balanced delicatessen products, aligning with the evolving demands of health-conscious consumers.

## Figures and Tables

**Table 1 animals-14-01980-t001:** Composition of raw materials used for making Mortadella (means ± DS).

	Pistachio	Cow Meat	Young Bull Meat	Pork Lard	Pork Meat
pH	ND	5.61 ± 0.13	5.55 ± 0.13	6.02 ± 0.03	5.40 ± 0.04
Moisture %	3.28 ± 0.15	71.48 ± 0.40	72.64 ± 0.17	20.19 ± 0.07	78.95 ± 0.79
Protein, %	22.16 ± 0.61	15.33 ± 0.15	16.60 ± 0.24	5.09 ± 0.06	11.62 ± 0.47
Fat, %	56.95 ± 0.80	12.08 ± 0.30	9.86 ± 0.17	72.67 ± 0.07	8.24 ± 0.23
Ash, %	2.50 ± 0.10	0.92 ± 0.04	0.76 ± 0.06	0.35 ± 0.08	0.87 ± 0.02
POV, mEq O_2_/kg fat	ND	0.30 ± 0.00	0.23 ± 0.03	0.85 ± 0.03	0.28 ± 0.01

POV = peroxide value.

**Table 2 animals-14-01980-t002:** Physical and chemical parameters of mixtures of Mortadella.

	Cow Mixture	Young Bull Mixture	Pork Mixture	SEM	Significance
pH	5.72 ^b^	5.68 ^b^	6.08 ^a^	0.071	0.0134
a_w_	0.843 ^b^	0.866 ^a^	0.854 ^ab^	0.004	0.0226
Moisture %	53.22 ^c^	56.44 ^b^	61.72 ^a^	0.482	<0.0001
Protein, %	15.40 ^b^	17.17 ^a^	13.07 ^c^	0.224	<0.0001
Fat, %	26.87 ^a^	22.61 ^b^	20.91 ^c^	0.205	<0.0001
Ash, %	3.22 ^a^	2.70 ^c^	2.93 ^b^	0.045	0.0005
Carbohydrates, %	1.28 ^ab^	1.08 ^b^	1.36 ^a^	0.080	0.0112
Caloric value, Kcal/100 g	308.6 ^a^	276.5 ^b^	245.9 ^c^	2.832	<0.0001
POV, mEq O_2_/kg fat	0.554	0.539	0.543	0.010	0.5516
L*, Lightness	51.84 ^c^	55.16 ^b^	66.82 ^a^	0.591	<0.0001
a*, Red index	12.07 ^a^	10.92 ^a^	6.51 ^b^	0.434	0.0002
b*, Yellow index	12.09 ^a^	11.85 ^a^	8.93 ^b^	0.480	0.0061
Chroma	17.08 ^a^	16.12 ^a^	11.06 ^b^	0.577	0.0007
Hue	45.02 ^b^	47.27 ^b^	54.07 ^a^	1.324	0.0070

The results indicate mean values of three measurements performed on each of samples. SEM = standard error of the means; POV = peroxide value. On horizontal rows: a, b, and c = *p* ≤ 0.05.

**Table 3 animals-14-01980-t003:** Physical and chemical parameters of Mortadella.

	Cow Mortadella	Young Bull Mortadella	Pork Mortadella	SEM	Significance
pH	5.74 ^c^	6.16 ^b^	6.70 ^a^	0.059	<0.0001
a_w_	0.940	0.943	0.941	0.007	0.9723
Moisture %	50.99 ^c^	54.59 ^b^	59.68 ^a^	0.242	<0.0001
Protein, %	16.92 ^b^	17.84 ^a^	14.07 ^c^	0.133	<0.0001
Fat, %	27.59 ^a^	23.93 ^b^	21.96 ^c^	0.147	<0.0001
Ash, %	2.66 ^a^	2.11 ^c^	2.48 ^b^	0.024	<0.0001
Carbohydrates, %	1.83 ^a^	1.52 ^b^	1.80 ^a^	0.041	0.0005
Caloric value, Kcal/100 g	323.4 ^a^	292.9 ^b^	261.1 ^c^	1.673	<0.0001
POV, mEq O_2_/kg fat	0.275	0.249	0.267	0.007	0.0786
L*, Lightness	54.37 ^c^	53.16 ^b^	67.09 ^a^	0.218	<0.0001
a*, Red index	19.95 ^a^	19.80 ^a^	14.89 ^b^	0.250	<0.0001
b*, Yellow index	11.66 ^a^	11.84 ^a^	8.71 ^b^	0.080	<0.0001
Chroma	23.11 ^a^	23.07 ^a^	17.25 ^b^	0.220	<0.0001
Hue	30.30	30.88	30.34	0.464	0.6381
Hardness N/mm^2^	0.352 ^a^	0.306 ^b^	0.304 ^b^	0.009	0.0132

The results indicate mean values of three measurements performed on each of samples. SEM = standard error of the means; POV = peroxide value. On horizontal rows: a, b, and c = *p* ≤ 0.05.

**Table 4 animals-14-01980-t004:** Fatty acids in raw materials used for making Mortadella (means ± DS).

	Pistachio	Cow Meat	Young Bull Meat	Pork Lard	Pork Meat
Total fatty acids (FA), %	51.06 ± 0.34	10.83 ± 0.44	8.84 ± 0.03	65.16 ± 1.12	7.39 ± 0.28
SFA, % FA	13.68 ± 0.35	50.99 ± 1.65	47.14 ± 1.66	35.96 ± 1.18	42.54 ± 1.45
MUFA, % FA	68.85 ± 1.38	41.38 ± 1.14	48.44 ± 1.09	48.34 ± 0.99	47.83 ± 0.98
PUFA, % FA	17.47 ± 0.36	7.27 ± 0.28	4.23 ± 0.20	15.70 ± 0.40	9.59 ± 0.28
ω3, % FA	0.12 ± 0.00	1.99 ± 0.02	0.66 ± 0.04	1.89 ± 0.02	1.34 ± 0.01
ω6, % FA	17.33 ± 0.43	5.15 ± 0.08	3.31 ± 0.07	15.11 ± 0.23	9.35 ± 0.11

SFA = saturated FA; MUFA = monounsaturated FA; PUFA = polyunsaturated FA.

**Table 5 animals-14-01980-t005:** Saturated fatty acids (% of total FA) of mortadella.

	Cow Mortadella	Young Bull Mortadella	Pork Mortadella	SEM	Significance
C8:0	0.018 ^b^	0.019 ^b^	0.028 ^a^	0.001	<0.0001
C10:0	0.068 ^b^	0.066 ^b^	0.115 ^a^	0.002	<0.0001
C12:0	0.079 ^b^	0.085 ^b^	0.168 ^a^	0.003	<0.0001
C14:0	2.24 ^a^	2.38 ^a^	2.01 ^b^	0.064	0.0170
C15:0	0.389 ^b^	0.447 ^a^	0.056 ^c^	0.013	<0.0001
C16:0	24.61 ^b^	25.46 ^a^	25.37 ^a^	0.288	0.0210
C17:0	1.05 ^a^	1.01 ^a^	0.71 ^b^	0.027	0.0002
C18:0	11.79	11.52	11.72	0.337	0.8488
C20:0	0.118 ^b^	0.097 ^b^	0.140 ^a^	0.009	<0.0001
C22:0	0.047	0.040	0.040	0.002	0.0609

The results indicate mean values of three measurements performed on each sample. SEM = standard error of the means; FA = fatty acids. On horizontal rows: a, b, and c = *p* ≤ 0.05.

**Table 6 animals-14-01980-t006:** Monounsaturated fatty acids (% of total FA) of mortadella.

	Cow Mortadella	Young Bull Mortadella	Pork Mortadella	SEM	Significance
C12:1	0.085 ^a^	0.064 ^b^	0.027 ^c^	0.002	0.0057
C14:1	0.282 ^a^	0.260 ^a^	0.071 ^b^	0.008	<0.0001
C15:1	0.384 ^a^	0.348 ^a^	0.180 ^b^	0.019	<0.0001
C16:1	3.51 ^a^	3.72 ^a^	2.65 ^b^	0.096	0.0005
C17:1	0.696 ^a^	0.781 ^a^	0.340 ^b^	0.018	<0.0001
C18:1 c9 OA	42.01 ^b^	41.73 ^b^	44.51 ^a^	0.494	0.0136
C18:1 t11 TVA	1.50 ^a^	0.90 ^b^	0.36 ^c^	0.030	<0.0001
Other C18:1	0.948 ^a^	0.655 ^b^	0.414 ^c^	0.020	0.0039
C20:1 n11	0.030	0.019	0.029	0.007	0.0882

The results indicate mean values of three measurements performed on each sample. SEM = standard error of the means; FA = fatty acids; OA = oleic acid; TVA = trans vaccenic acid. On horizontal rows: a, b, and c = *p* ≤ 0.05.

**Table 7 animals-14-01980-t007:** Polyunsaturated fatty acids (% of total FA) of mortadella.

	Cow Mortadella	Young Bull Mortadella	Pork Mortadella	SEM	Significance
Other C18:2	0.451 ^a^	0.165 ^b^	0.132 ^b^	0.028	<0.0001
C18:2 ω6 LA	7.46 ^b^	7.99 ^a^	7.94 ^a^	0.090	0.0109
CLA C18:2 c9 t11 RA	0.317 ^a^	0.231 ^b^	0.091 ^c^	0.017	<0.0001
Other CLA^c^ isomers	0.095 ^a^	0.082 ^a^	0.049 ^b^	0.002	<0.0001
C18:3 ω3 ALA	1.37 ^b^	1.01 ^c^	1.56 ^a^	0.038	0.0002
C18:3 ω6 GLA	0.060 ^a^	0.044 ^b^	0.024 ^c^	0.001	<0.0001
C20:2 ω6	0.067 ^a^	0.053 ^b^	0.062 ^a^	0.002	0.0034
C20:3 ω3	0.018	0.017	0.017	0.001	0.6778
C20:3 ω6 DGLA	0.306	0.316	0.297	0.009	0.3827
C20:4 ω6 AA	0.212 ^c^	0.250 ^b^	0.289 ^a^	0.007	0.0009
C20:5 ω3 EPA	0.052 ^b^	0.070 ^a^	0.047 ^b^	0.002	0.0001
C22:2 ω6	0.087	0.088	0.085	0.002	0.3843
C22:5 ω3 DPA	0.141 ^b^	0.076 ^c^	0.167 ^a^	0.004	<0.0001
C22:6 ω3 DHA	0.104 ^b^	0.068 ^c^	0.152 ^a^	0.003	<0.0001

The results indicate mean values of three measurements performed on each sample. SEM = standard error of the means; FA = fatty acids; LA = linoleic acid. RA = rumenic acid. CLA = conjugated linoleic acid. ALA = α-linolenic acid. GLA = γ-linolenic acid. DGLA = Diomo-γ-linolenic acid. AA = arachidonic acid; EPA = eicosapentaenoic acid; DPA = docosapentaenoic acid; DHA = docosahexaenoic acid. On horizontal rows: a, b, and c = *p* ≤ 0.05.

**Table 8 animals-14-01980-t008:** Fatty acid profile (% of total FA) and health indexes of mortadella.

	Cow Mortadella	Young Bull Mortadella	Pork Mortadella	SEM	Significance
Total FA, %	24.75 ^a^	21.46 ^b^	19.69 ^c^	0.324	0.0123
SFA	39.82	41.13	40.36	0.735	0.4901
MUFA	49.36	48.41	48.56	0.668	0.5876
PUFA	10.73	10.39	11.05	0.175	0.0966
PUFA/SFA	0.270	0.253	0.274	0.020	0.1058
ω6	9.50 ^b^	9.71 ^b^	10.23 ^a^	0.148	0.0325
ω3	1.68 ^b^	1.24 ^c^	1.94 ^a^	0.047	0.0001
ω6/ω3	5.65 ^b^	7.83 ^a^	5.27 ^b^	0.088	<0.0001
AI	0.550 ^c^	0.596 ^a^	0.562 ^b^	0.001	<0.0001
TI	1.10 ^b^	1.20 ^a^	1.10 ^b^	0.002	<0.0001

The results indicate mean values of three measurements performed on each sample. SEM = standard error of the means; FA = fatty acids; SFA = saturated FA; MUFA = monounsaturated FA; PUFA = polyunsaturated FA; AI = atherogenic index; TI = thrombogenic index. On horizontal rows: a, b, and c = *p* ≤ 0.05.

**Table 9 animals-14-01980-t009:** Sensorial profile of mortadella.

	Cow Mortadella	Young Bull Mortadella	Pork Mortadella	SEM	Significance
Color intensity	7.11 ^a^	7.28 ^a^	2.09 ^b^	0.382	<0.0001
Color homogeneity	6.09 ^a^	6.26 ^a^	4.77 ^b^	0.451	0.0419
Adhesion of fat cubes	6.15 ^a^	6.60 ^a^	3.87 ^b^	0.788	0.0540
Presence of gelatin pockets	0.78	0.85	1.45	0.521	0.1074
Odor intensity	5.65	5.65	4.90	0.505	0.4934
Odor persistence	5.03 ^a^	4.87 ^ab^	3.53 ^b^	0.516	0.0316
Spicy odor	2.94 ^b^	4.59 ^a^	2.38 ^b^	0.426	0.0005
Unpleasant odors	0.23 ^b^	0.33 ^b^	0.96 ^a^	0.196	0.0343
Salty	4.79	4.35	3.49	0.445	0.1395
Sweet	1.69	1.82	1.86	0.342	0.9351
Sour	3.68 ^a^	3.19 ^a^	2.46 ^b^	0.238	0.0069
Bitter	0.30	0.62	0.37	0.133	0.2290
Spicy	1.44	1.91	1.36	0.436	0.2933
Tenderness	4.33	4.36	6.15	0.638	0.0657
Chewability	3.86	3.64	2.99	0.488	0.4398
Fatness	3.13	4.16	4.08	0.384	0.1368
Overall acceptability	6.05	6.52	5.53	0.593	0.6076

The results indicate mean values on a hedonic scale of 10 cm divided into 10 hedonic points (0 = absent; 10 = intense). SEM = standard error of the means. On horizontal rows: a and b = *p* ≤ 0.05.

## Data Availability

All data included in this study are available upon request by contacting the corresponding author.

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
