# Peer review of "Traits of Mortadella from Meat of Different Commercial Categories of Indigenous Dairy Cattle"

_animals, 2024, doi:10.3390/ani14131980_

Round 1

Reviewer 1 Report

Comments and Suggestions for Authors

The review of the manuscript (ID: animals- 3059399) entitled: “Traits of Mortadella from Meat of Different Commercial Categories of Indigenous Dairy Cattle.”

Dear Authors,

with great interest I read the scientific article, which is correctly written and contributes significantly to the current state of knowledge information in terms of the physicochemical traits, fatty acid profile, and sensory properties of mortadella made with the meat of Cinisara young bulls and cows raised under feeding systems based on the use of natural resources, which is extremely important in sustainable farming and food production. As written by the authors, to obtain healthier animal products, recently explore alternative meat raw materials, from animals varied feeding systems, yielding products with healthier nutritional profile. This presents economic opportunities, especially for farms in marginal areas with non-specialized breeds.

The introduction to the topic is short and in my opinion it requires supplementation. In the paragraph about Cinisar's cows, lines 67-69, add the positive aspects of the meat from those cows that decided to use it in this experiment.

In the paragraph about mortadella, lines 70 - 75, describe in more detail (not in one sentence) why it was decided to add other types of meat and additives, and what are the effects of consuming mortadella products that contain up to 30% fat with a predominance of saturated and monounsaturated fatty acids.

The description of the course of the experiment is clear and clearly described. A detailed methodology allows other scientists to repeat or continue their research in this area.

Correctly selected methodologies and tools allowed the authors to present the results in an interesting way and to discuss them. The conclusions consistent with the evidence and arguments were presented and they are addressing the main goal.

I recommend the work for publication in the journal Animals.

Author Response

Dear Reviewer,

Thank you for your attention in reviewing our manuscript and for your comments which will allow us to improve its quality. We have responded to your indications by highlighting the changes made in the text.

Best regards

The authors

Comments 1: The introduction to the topic is short and in my opinion it requires supplementation. In the paragraph about Cinisar's cows, lines 67-69, add the positive aspects of the meat from those cows that decided to use it in this experiment.

Response 1: Thank you for your comment.

We add the following phrase “Furthermore, experimental tests conducted on calves raised on pasture and subsequently finished in the stable, according to the traditional breeding system, recorded valuable characteristics of the meat in terms of iron content, vitamin E, and the high concentration of conjugated linoleic acid (CLA). This is important in the prevention of cardiovascular diseases, while the lipid and cholesterol contents were significantly lower in muscle tissues compared to other breeds with a greater aptitude for meat production [22].”

Comments 2: In the paragraph about mortadella, lines 70 - 75, describe in more detail (not in one sentence) why it was decided to add other types of meat and additives, and what are the effects of consuming mortadella products that contain up to 30% fat with a predominance of saturated and monounsaturated fatty acids.

Response 2: Thank you for your comment.

Comments 3:  We rewrite the lines 70-79, as here reported:

Response 3: This product generally has high fat content, containing up to 30% fat, and presents unhealthy fatty acids characterized by prevalence of saturated and monounsaturated acids. To obtain Mortadella with a healthier nutritional profile, different studies have tested the use of alternative livestock such as cow, poultry, and donkey, and pork raised on natural resources, which, as is known, can improve the fatty acids profile in meat products by increasing polyunsaturated fatty acids content with important healthy effects for human consumers (11-21).

In this context, the present study provides information on the physicochemical traits, fatty acid profile, and sensory properties of lower fat content mortadella made with the meat of Cinisara young bulls and cows at the end of their productive life adding pork lard from the Nebrodi Black Pig, with the intent to also improve their fatty acid profiles, using animals raised on the prevalence use of natural resources.

Reviewer 2 Report

Comments and Suggestions for Authors

Both meat product reformulation and use of native breeds are very actual topics in meat industry. The article is well written; however, I have several suggestions and queries:

L40: I suggest to add the following keywords: "meat products", "reformulation"

L71: slaughter animals

L100-102: It would be good to generally specify the commercial names of additives - e.g. Foxtec 201/1 (phosphates), curing salt, mix of spices, etc.

L121: the pH was measured

L137: I understand using freeze-drying prior to determining fatty acid profile, but it doesn't seem such a good idea to partially remove water by sublimation before dry matter determination by gravimetry. Would it not interfere with the rate of evaporation, even if using the original weight before freeze-drying for dry-matter calculation?

L180-181: please provide the pH range

L194-196: It seems to me that the raw meat was rather low on crude protein, especially pork (less than 12% with 79% of water; Table 1). That is not much similar to the nutrition values in food databases, as you say.

L223-224: please join the paragraphs. A new paragraph may start on line 227.

L255: Table 1

L263: arachidic acid

The significance in Tables is P ≤ .05, while in the statistical analysis description it is the more usual P < .05

I suggest to put the small letters into superscript, it would also help with formatting in Table 9.

Table 9: I suggest to use "Pork mortadella" as in the rest of tables, or "Control (pork) mortadella" everywhere.

Author Response

Dear Reviewer,

Thank you for your attention in reviewing our manuscript and for your comments which will allow us to improve its quality. We have responded to your indications by highlighting the changes made in the text.

Best regards

The authors

Comments 1:  Both meat product reformulation and use of native breeds are very actual topics in meat industry. The article is well written; however, I have several suggestions and queries:

L40: I suggest to add the following keywords: "meat products", "reformulation"

Response 1: Thank you. Done

Comments 2: L71: slaughter animals

Response 2: Thank you. Done

Comments 3:  L100-102: It would be good to generally specify the commercial names of additives - e.g. Foxtec 201/1 (phosphates), curing salt, mix of spices, etc.

Response 3: Thank you. Done

Comments 4: L121: the pH was measured

Response 4: Thank you. Done

Comments 5: L137: I understand using freeze-drying prior to determining fatty acid profile, but it doesn't seem such a good idea to partially remove water by sublimation before dry matter determination by gravimetry. Would it not interfere with the rate of evaporation, even if using the original weight before freeze-drying for dry-matter calculation?

Response 5: Thank you for your comments. In another experiment, we tested that the use of freeze-drying for dry matter determination doesn't influence the dry matter content.

Comments 6:  L180-181: please provide the pH range

Response 6: Thank you.  We rewrite lines 180-182 as following: During the production process, each type of meat (Tables 1, 2, 3) exhibited pH values  ideal for the transformation into Mortadella, achieving pH values (5.74-6.16) slightly lower than those typically observed in this product (6.00-6.71) [14,15,19,21,28,29].

Comments 7: L194-196: It seems to me that the raw meat was rather low on crude protein, especially pork (less than 12% with 79% of water; Table 1). That is not much similar to the nutrition values in food databases, as you say.

Response 7: Thanks for your comments. In table 1 the data are expressed as % dry matter, while in the database as % fresh meat. Calculating the data in the database on dry matter, the values are comparable.

Comments 8: L223-224: please join the paragraphs. A new paragraph may start on line 227.

Response 8: Thank you. Done

Comments 9: L255: Table 1

Response 9: Thanks you. it was checked and the table is the 4

Comments 10: L263: arachidic acid

Response 10: Thank you. Done

Comments 11:  The significance in Tables is P ≤ .05, while in the statistical analysis description it is the more usual P < .05

Response 11: Thanks. We corrected in material and methods.

Comments 12: I suggest to put the small letters into superscript, it would also help with formatting in Table 9.

Response 12: Thank you. Done

Comments 13:  Table 9: I suggest to use "Pork mortadella" as in the rest of tables, or "Control (pork) mortadella" everywhere.

Response 13: Thank you. done